# No Interaction between Polymorphisms Related to Vitamin A Metabolism and Vitamin A Intake in Relation to Colorectal Cancer in a Prospective Danish Cohort

**DOI:** 10.3390/nu11061428

**Published:** 2019-06-25

**Authors:** Vibeke Andersen, Ulrich Halekoh, Torsten Bohn, Anne Tjønneland, Ulla Vogel, Tine Iskov Kopp

**Affiliations:** 1Focused Research Unit for Molecular Diagnostic and Clinical Research, Institute of Regional Health Research-Center Sønderjylland, Hospital of Southern Jutland, 6200 Aabenraa, Denmark; 2Institute of Molecular Medicine, University of Southern Denmark, 5000 Odense, Denmark; 3Institute of Regional Health Research, University of Southern Denmark, 5000 Odense, Denmark; 4Institute of Public Health, Unit of Epidemiology, Biostatistics and Biodemography, University of Southern Denmark, 5000 Odense, Denmark; uhalekoh@health.sdu.dk; 5Luxembourg Institute of Health, Department of Population Health, 1445 Strassen, Luxembourg; Torsten.Bohn@lih.lu; 6Danish Cancer Society Research Center, 2100 Copenhagen, Denmark; annet@cancer.dk (A.T.); tine.iskov.kopp@regionh.dk (T.I.K.); 7Department of Public Health, Faculty of Health and Medical Sciences, University of Copenhagen, 2200 Copenhagen, Denmark; 8National Research Centre for the Working Environment, 2100 Copenhagen, Denmark; UBV@nfa.dk; 9The Danish Multiple Sclerosis Registry, Department of Neurology, Copenhagen University Hospital, Rigshospitalet, 2100 Copenhagen, Denmark

**Keywords:** gene-environment interaction, diet, immune system, candidate gene, pro-vitamin A-carotenoids, Vitamin A, retinol, Western-style diet

## Abstract

Although vitamin A is essential for gut immune cell trafficking (paramount for the intestinal immune system), epidemiological studies on the role of vitamin A in colorectal cancer (CRC) aetiology are conflicting. By using functional polymorphisms, gene–environment (GxE) interaction analyses may identify the biological effects (or “mechanism of action”) of environmental factors on CRC aetiology. Potential interactions between dietary or supplemental vitamin A intake and genetic variation in the vitamin A metabolic pathway genes related to risk of CRC were studied. We used a nested case-cohort design within the Danish “Diet, Cancer and Health” cohort, with prospectively collected lifestyle information from 57,053 participants, and the Cox proportional hazard models and likelihood ratio test. No statistically significant associations between the selected polymorphisms and CRC, and no statistically significant interactions between vitamin A intake and the polymorphisms were found. In conclusion, no support of an involvement of vitamin A in CRC aetiology was found.

## 1. Introduction

Colorectal cancer (CRC) has a high impact on human health with a lifetime risk in Western European and North American populations of around 5% [1]. CRC is the third most common cancer worldwide with more than 1.8 million new cases in 2018 and increasing incidence [1]. A significant part of the risk has been attributed to the Western life style [2], where a high intake of red and processed meat and alcohol use are linked to a high risk of CRC, and intake of whole grains and dairy products to a low risk of CRC [3,4,5]. With the goal of reducing the number of CRC patients, it is of major importance to identify individuals at high risk of CRC as well as the risk factors involved.

From biological considerations, vitamin A and pro-vitamin A carotenoids are likely to affect colorectal carcinogenesis. Vitamin A is a group of unsaturated nutritional organic compounds that includes retinal, retinol, retinoic acid (RA), and pro-vitamin A carotenoids (e.g., β-carotene, α-carotene, β-cryptoxanthin). Vitamin A sources from food include mainly fat-soluble retinol from animal sources (e.g., liver) and green- or yellow-coloured carotenoids from vegetables (e.g., leafy vegetables, carrots). In the intestine, RA (produced from retinol via two sequential oxidative steps) has been found to govern the regulation of T cells into Th1/Th2/Th17/Tregs pathways and the development of oral immune tolerance via induction of foxp3+ regulatory T cells [6]. In line with this, RA was found to be necessary for developing gut immune tolerance in response to certain microbes in an experimental human study [7], and RA-deficiency led to dysregulated T cell response and development of colitis and CRC in a mouse model [8]. Interestingly, RA is needed for the production of integrin α4β7 necessary for homing of T cells [9], which is an important initial step in the establishment of gut inflammation, and lack of β7 inhibited the growth of intestinal tumours in an animal model [10]. Additionally, pro-vitamin A carotenoids have been found to have antioxidant and anti-inflammatory properties mediated via e.g., activation of the nuclear factor (erythroid-derived 2)-like 2 (Nrf-2) pathway [11] and inhibition of the nuclear factor kappa-B (NF-kB) pathway, respectively, in in vitro studies (reviewed in [12]). Finally, RA can interact with the nuclear receptors RAR/RXR, which are related to immune responses and inflammation-related pathways [13]. These data are supported by human and animal studies; low serum retinol levels were identified as a predictor of poor survival in CRC patients [14], and vitamin A deficiency was associated with a higher rate of CRC development in an animal model [8]. However, despite the biologically plausible effect of vitamin A intake on CRC risk, the epidemiological evidence is scarce and conflicting [15,16,17,18]. Individual variability in the bioavailability due to e.g., genetics may, furthermore, complicate the investigations [19].

CRC is heterogeneous; hence many factors are involved in the disease aetiology. These factors may not be present in every individual patient developing CRC, and their impact may vary among individuals. Thus, even identification of important risk factors in subgroups may be difficult in epidemiological studies. Gene–environment interaction analyses assume that the genetic variants are randomly distributed during the gamete formation. Such analyses may capture risk factors present in specific subgroups of CRC patients that may not easily be captured by epidemiological studies, because an interaction signifies the involvement of both the studied gene and the environmental risk factor in the disease pathway [20,21,22,23,24].

Based on the scarce evidence on the role of vitamin A in CRC carcinogenesis, we decided to investigate potential interactions between vitamin A intake and gene variants related to the vitamin A metabolic pathway. We used a nested case-cohort design within the Danish “Diet, Cancer and Health” (DCH) cohort with prospectively collected lifestyle information encompassing 57,053 participants, of which 1038 cases that developed CRC were compared to 1857 controls. Vitamin A intake was analysed as total intake (dietary and supplements combined), dietary intake (retinol and β-carotene as a weighted sum), and supplements separately. We also evaluated potential effects of retinol (mainly from meat and meat products) and β-carotene (mainly from plant foods) separately. We selected functional variants in genes involved in vitamin A metabolism.

## 2. Materials and Methods

### 2.1. Subjects

As previously described [25], the DCH Study is a Danish cohort study designed to investigate the relation between diet, lifestyle, and cancer risk. The cohort consists of 57,053 persons, recruited between December 1993 and May 1997. All the subjects were born in Denmark, and the individuals were 50 to 64 years of age and had no previous cancers reported. Blood samples, anthropometric measures, and questionnaire data on diet and lifestyle were collected at study entry.

### 2.2. Follow-Up and Endpoints

As previously described [20,21,22,26,27], the present study used a nested case-cohort design. Follow-up was based on the Danish population-based Cancer Registry. Between 1994 and December 31, 2009, 1038 CRC cases were diagnosed. A sub-cohort of 1857 people was randomly selected as controls within the full cohort at time of entry in agreement with the case-cohort study design [28], and thus without respect to time and disease status. Due to the choice of design with a priori sampling of the sub-cohort, 28 persons were both cases and sub-cohort, and these persons were kept in the analyses. Consequently, 1038 CRC cases and 1857 sub-cohort members were analysed.

### 2.3. Dietary and Lifestyle Questionnaire

Information on diet, lifestyle, weight, height, medical history, environmental exposures, and socio-economic factors were collected at enrolment using questionnaires and interviews and has been described in detail elsewhere [21,29,30]. In short, the validated food-frequency questionnaire, assessed dietary intake in 12 categories of predefined responses, including 68 food items, ranking from ‘never’ to ‘eight times or more per day’. A section on the intake of dietary supplements included open-ended questions on brands and doses, and categorical questions on frequency of intake and its duration (number of months during the last year) and whether they had taken the supplement in question within the last month. Information on the contents of micronutrients in the different brands of dietary supplement was obtained from producers or distributors of the specific products. For each participant we calculated average daily intake of specific foods and nutrients by means of the software program Food Calc (Copenhagen University, Copenhagen, Denmark) [31], using population-specific standardized recipes and sex-specific portion sizes. The two forms of vitamin A, retinol and β-carotene, were examined as a weighted sum, where β-carotene was allotted one sixth of the vitamin A-activity compared to retinol [30]. The vitamin A from supplements was calculated as previously described [30]. The Pearson correlation coefficients for nutrient intake (adjusted for energy intake) between the food-frequency questionnaire and two times 7 days of weighted records was 0.45 for vitamin A. Red and processed meat was calculated by combining intake of fresh and minced beef, veal, pork, lamb, offal, bacon, smoked or cooked ham, other cold cuts, salami, frankfurter, Cumberland sausage, and liver pâté. Total dietary fiber was estimated by the method of the Association of Official Analytical Chemists [32], which included lignin and resistant starch. Fiber intake was calculated by multiplying the frequency of consumption of relevant foods (i.e., fruit, vegetables, grains, and leguminous fruit) by their fiber content as determined from national databases of food content as described earlier [21,29,30]. For fruits, only the intake of fresh fruit was examined, whereas intake of vegetables also included estimated contributions from food recipes. Intake of alcohol was inferred from the food-frequency questionnaire and life-style questionnaire as described earlier [33]. Abstainers were defined as those who reported no intake of alcohol on the food-frequency questionnaire, and no drinking occasions on the lifestyle questionnaire. Smoking status was classified as never, past, or current. Persons smoking at least 1 (one?) cigarette daily during the last year were classified as smokers. Non-steroidal anti-inflammatory drug (NSAID) use (“Aspirin”, “Ibuprofen”, or “Other pain relievers”) was assessed as ≥ 2 pills per month during one year at baseline. Use of hormone replacement therapy (HRT) among women was assessed as current, former, or never user.

### 2.4. Genotyping and Selection of Polymorphisms

The polymorphisms were chosen based on their biological function. Promising polymorphisms with known functionality or associated with biological effects suggesting functionality or linkage with functional polymorphisms, and with a reasonable minor allele frequency to study gene–environment interactions, were selected. Buffy coat preparations were stored at minus 150 °C until use. DNA was extracted as described [34]. The DNA was genotyped by LGC KBioscience (Hoddesdon, UK) by polymerase chain reaction (PCR)-based KASP™ genotyping assay (http://www.lgcgenomics.com/). To confirm reproducibility, genotyping was repeated for 10% of the samples, yielding 100% identity.

### 2.5. Statistics

Incidence rate ratios (IRR) and 95% Confidence Intervals (CI) were based on a Cox proportional hazard model fitted to the age at the event of CRC according to the principles for analysis of case-cohort studies [28], using the approach of Prentice and Langholz [35]. The main explanatory variables were the genotypes. All models were adjusted for baseline values of risk factors for CRC: body mass index (BMI) (kg/m^2^, continuous), use of hormone replacement therapy (HRT), (never/past/current, among women), intake of dietary fibre (g/day, continuous), red and processed meat intake (g/day, continuous), energy intake (kJ/day), NSAID use (yes/no), and smoking status (never/past/current). Cereal, fibre, fruit, and vegetable consumption were also entered linearly as continuous covariates. All analyses were stratified by gender to ensure that baseline (underlying) hazards were gender specific.

No recessive effects were found. In order to maximize the statistical power for the interactions analyses, the genotypes were, therefore, combined, assuming a dominant model. In the interaction analyses for vitamin A intake and polymorphisms, we present two analyses: in one analysis the vitamin A intake was used as a numeric variable, and in the other, vitamin A intake was entered in the model as a three-level categorical variable, defined via tertile cut-points derived from the empirical distribution of the whole population. Deviation from the Hardy–Weinberg equilibrium in the comparison group was assessed using a Chi-square test. All analyses were performed using the survival package (Terry M. Therneau, version 2.42.4, (Mayo Clinic, Rochester, MN, USA)) of the statistical computational environment R, version 3.5.1 ((R Foundation for Statistical Computing, Vienna, Austria)). A *p* < 0.05 (2-sided) was considered to indicate a statistically significant test result.

## 3. Results

### 3.1. Baseline Characteristics

Figure 1 shows the flowchart of the study participants. Included in the analysis were 1038 cases and a subcohort of 1857 control participants.

Table 1 shows the baseline characteristics of 1038 CRC cases and 1857 sub-cohort members including CRC risk factors as previously studied [20,21,22,24,27,36,37]. There were no associations between vitamin A intakes and risk of CRC. Among the controls, the genotype distributions of the studied polymorphisms were in Hardy–Weinberg equilibrium (results not shown).

### 3.2. Associations between Polymorphisms and Colorectal Cancer (CRC)

Table 2 shows the studied polymorphisms (rs1667255 near *TTR*, *FFAR4* rs10882272, rs4889286 near *BCO1*, *BCO1* rs12934922, rs6564851 near *BCO1*, *RARB* rs6800566, *RARB* rs13070407, *ABCA1* rs2791952, *FABP2* rs1799883). The polymorphisms were selected based on known or suggested functional effects such as being involved in the metabolism, transport, or cellular uptake of retinoids.

Table 3 shows the crude and adjusted associations between the polymorphisms and CRC. No statistically significant associations were found. In order to maximize the statistical power for the gene–vitamin A interaction analyses, the genotypes were combined, assuming a dominant model.

### 3.3. Interactions between Polymorphisms and Vitamin A Intake

Table 4 shows the results from the interaction analysis between the total dietary intake of vitamin A, β-carotene, and retinol and the targeted polymorphisms, and Table 5 shows the results from the tertile analysis of the polymorphisms and total dietary intake of A vitamin, β-carotene, and retinol. There were no statistically significant interactions between the polymorphisms and total intake of vitamin A, β-carotene, and retinol. However, weak associations between *BCO1* rs4889286 and *BCO1* rs6564851 and intake of β-carotene and risk of CRC were seen so that homozygotes of the most common genotype were associated with 20% and 21% increased risk of CRC, respectively (IRR*_BCO1_*
_rs4889286_ = 1.20, 95% CI 0.97–1.49, *P*_for interaction_ = 0.08; IRR*_BCO1_*
_rs6564851_ = 1.21, 95%: 0.98–1.50, *P*_for interaction_ = 0.09) (Table 3). In tertile analyses, variant carriage of these two genotypes exhibited a dose-dependent decrease in CRC risk; that is, the risk decreased with increasing intake of β-carotene (Table 5). Similar patterns were found for the other polymorphisms and β-carotene. The opposite was observed for retinol, where the risk of CRC increased with increasing intake of retinol. No statistically significant interactions were found between polymorphisms and vitamin A from dietary intake or from supplements, respectively (data not shown), in relation to risk of CRC.

## 4. Discussion

This study investigated potential associations between polymorphisms of genes involved in the vitamin A metabolic pathway and CRC and, furthermore, potential interactions between these polymorphisms and dietary intake of vitamin A in relation to CRC risk using the large prospective Danish “Diet, Cancer and Health” cohort. We found no statistically significant associations of the polymorphisms with risk of CRC, and no statistically significant interactions between the selected polymorphisms and intake of vitamin A in relation to CRC risk. When evaluating β-carotene and retinol separately, we did find an indication of opposite CRC risks independent of genotype, where β-carotene was associated with dose-dependent decreased CRC risks, and retinol was associated with dose-dependent increased CRC risks. These effects are most likely driven by plant foods and meat, respectively, and not vitamin A, due to the lack of effect of the polymorphisms.

The epidemiological evidence for the relation between vitamin A and CRC aetiology is scarce and conflicting. First, a large meta-analysis from 2010 found no association between dietary intake of vitamin A from food and supplements and CRC (relative risks (RR) = 0.88; 95% CI: 0.76,1.02 for >4000 vs. ≤1000 μg/day). The study included thirteen prospective cohort studies from North America and Europe in a pooled analysis comprising 676,141 men and women, including 5454 CRC cases [17]. Conversely, two prospective studies from 2012 and 2014 found inverse associations of pre-diagnostic blood concentrations of RA with CRC risk. In the first study, nested within the European Prospective Investigation into Cancer and Nutrition (EPIC), plasma concentrations of RA and dietary consumption of RA were determined in 898 colon cancer cases, 501 rectal cancer cases, and 1399 matched controls. An inverse association was observed between high pre-diagnostic plasma RA concentration and a low risk of colon cancer (IRR for the highest quartile = 0.63 (0.46–0.87), *p*_trend_ = 0.01) [15]. The other study evaluated the association of serum levels of eight antioxidant nutrients, including β-carotene, among postmenopausal women from a subsample of the Women’s Health Initiative in relation to CRC risk using repeated measurements. Among 5477 women with baseline serum antioxidant values, 88 incident cases of CRC were identified over a median follow-up time of 12 years. The average serum level of β-carotene was inversely associated with lowered risk of CRC (HRs for highest vs. lowest tertile 0.54 (0.31–0.96)) [16]. A retrospective case-control study, the Japanese Fukuoka CRC study, published in 2012, found that intake of vitamin A (retinol-equivalent) was inversely related to CRC risk in women, but positively related in men [18]. Thus, these studies do not support the notion that a high vitamin A intake protects against CRC. Our genetic study found no support for a significant role of vitamin A in the etiology of CRC.

Furthermore, vitamin A and, especially, carotenoid absorption from supplements, may be hampered by being less efficient compared to dietary intake, especially from formulations not containing many lipids [46]. Therefore, in addition to analysing the total vitamin A intake (i.e., from both diet and supplements), we analysed the data for dietary intake and intake from supplements separately. No statistically significant interactions between the polymorphisms and the intake of vitamin A from dietary sources or from supplements were found. Similarly, inconsistent associations were found for CRC risk and supplemental vitamin A in a recent review [47]. However, the power in the present study to perform such analysis was limited.

Advantages and limitations with the study design have been described in previous studies [20,21,22,26,27,48]. The main advantage of this study is the prospective study design including the collection of dietary and lifestyle factors before diagnosis which eliminates the risk of recall bias. This design has previously proven its value by the identification of GxE interactions between e.g., meat and pattern recognition receptors [20,21]. Changes in dietary and life-style habits during follow-up are possible and, if present, would result in lower power to detect real differences between cases and the comparison group. The “Diet, Cancer and Health” cohort is relatively homogenous, reducing population-specific differences in genetics and dietary patterns seen in larger multicentre studies. The disadvantage of this prospective study is the limited power to study gene–environment interactions. Residual confounding of lifestyle factors might be present; however, the study design using GxE interaction analysis with polymorphisms selected on biological basis strongly reduces the risk of bias. Alcohol intake could potentially have affected the results. In a case-control study, an association between the serum retinol level and head and neck cancer was found that was not present among regular alcohol consumers [39]. A high number of alcohol drinkers in our study may, therefore, result in a reduced ability to detect an interaction between carotenoids and CRC despite the adjustment for alcohol intake. Finally, the null-result of this study is dependent on several premises. First, the selected polymorphisms are either functional themselves or in linkage with functional polymorphisms involved in vitamin A metabolism. This criteria, however, seems to be fulfilled [19,38,39,40,41,42,43,45]. Next, the intake of vitamin A should be sufficiently distributed among the study participants to allow the evaluation of variable intakes. As shown in Table 1, the intake of vitamin A seems to be well-distributed among the participants. However, nearly all participants appeared to have a sufficient vitamin A intake, as the recommended intake of vitamin A is 700 and 900 RE/day for women and men, respectively, which was generally surpassed.

## 5. Conclusions

In conclusion, this large gene–environment interaction analysis, using a nested case-cohort design within the Danish “Diet, Cancer and Health” cohort with prospectively collected lifestyle information encompassing 57,053 participants, found no support of an involvement of vitamin A in CRC aetiology.

## Figures and Tables

**Figure 1 nutrients-11-01428-f001:**
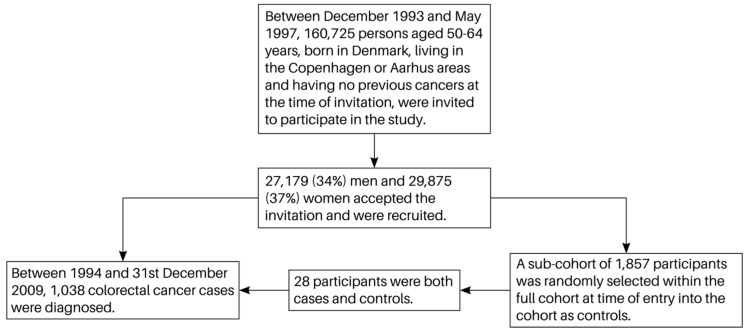
Flow chart of study participants.

**Table 1 nutrients-11-01428-t001:** Participant description.

Variable	Cases		Sub-Cohort		IRR (95% CI) ^1^
	*n* (%)	Median (5–95%) (Missing)	*n* (%)	Median (5–95%) (Missing)	
Total	1038 (100)		1857 (100)		
**Sex**					
Females	462 (45)		865 (47)		
Males	576 (55)		992 (53)		
Age at entry		58 (51–65)		56 (51–64)	
BMI (kg/m^2^)		26 (21–34) (3)		26 (21–33)	1.05 (1.01–1.10) ^5^
**Food intake**					
Alcohol (g/day) ^2^		15 (1–71)		14 (1–66)	1.03 (0.98–1.07) ^6^
Dietary fiber (g/day)		20 (11–33)		21 (11–34)	0.83 (0.65–1.08) ^7^
Red and processed meat (g/day)		112 (46–233)		109 (41–236)	1.01 (0.97–1.06) ^8^
Total energy (kJ/day)		9681 (6115–14,712) (4)		9633 (5922–14,820)	1.00 (1.00–1.00) ^9^
Fruits (g/day)		166 (24–493) (4)		176 (27–546)	0.98 (0.95–1.02) ^10^
Vegetables (g/day)		153 (46–367) (4)		163 (50–372)	1.03 (0.98–1.09) ^11^
Fruit and vegetables (g/day)		331 (98-796) (4)		350 (102–818)	1.00 (0.97–1.02) ^12^
**Vitamin A intake**					
Vitamin A (total) (RE/day)		1980 (759–4332) (0)		1992 (800–4398) (4)	0.98 (0.90–1.06) ^13^
Vitamin A (dietary) (RE/day)		1684 (697–4189) (0)		1693 (670–3966) (4)	1.03 (0.94–1.13)
β-Carotene (total) (µg/day)		2993 (707–12,424) (0)		3177 (761–12,837) (4)	0.99 (0.89–1.10) ^14^
Retinol (total) (µg/day)		1033 (285–2661) (0)		966 (272–2550) (4)	1.04 (0.91–1.18) ^15^
Vitamin A (supplements) (%, *n*)	455 (44)		882 (48) [4]		0.87 (0.74–1.03)
Vitamin A (supplements) ^3^ (RE/day)		624 (72–1011) (0)		640 (57–1500) (4)	
**Smoking status**					
Never	306 (29)		621 (33)		1.00 (ref.)
Past	322 (31)		536 (29)		1.12 (0.91–1.38)
Current	410 (39)		699 (38)		1.18 (0.97–1.44)
**NSAID use ^4^**					
No	716 (70)		1275 (69)		1.00 (ref.)
Yes	313 (30)		568 (31)		0.99 (0.84–1.18)
**HRT use among women**					
Never	279 (60)		455 (53)		1.00 (ref.)
Past	62 (13)		137 (16)		0.65 (0.45–0.92)
Current	121 (26)		273 (32)		0.70 (0.53–0.92)

Values are expressed as medians (5th and 95th percentiles) or as fractions (%). Number of missing observations in parenthesis. IRR, incidence rate ratio; CRC, colorectal cancer; CI, confidence interval; BMI, body mass index; NSAID, non-steroidal anti-inflammatory drug; HRT, hormone replacement therapy. ^1^ IRRs for CRC estimated by the Cox proportional hazards model mutually adjusted for all variables, with age as the underlying time axis, and stratified by gender, so that the underlying baseline hazards are gender specific. ^2^ Among current drinkers. ^3^ For those taking supplements. ^4^ NSAID use is defined as ≥2 pills per month for one year. ^5^ Risk estimate per 2 kg/m^2^ increment of BMI. ^6^ Risk estimate for the increment of 10 g alcohol per day. ^7^ Risk estimate for the increment of 10 g dietary fibers per day. ^8^ Risk estimate for the increment of 25 g red and processed meat per day. ^9^ Risk estimate for the increment of 1 kJ energy per day (incl alcohol). ^10^ Risk estimate for the increment of 50 fruits per day. ^11^ Risk estimate for the increment of 50 vegetables per day. ^12^ Risk estimate for the increment of 50 g fruits or vegetables per day. ^13^ Risk estimate for the increment of 1000 retinol equivalents (RE) per day. ^14^ Risk estimate for the increment of 4000 µg β-carotene per day. ^15^ Risk estimate for the increment of 1000 µg retinol per day.

**Table 2 nutrients-11-01428-t002:** Suggested biological effects of the selected polymorphisms.

Gene	Rs-Number	MAF	Function/Effect of Polymorphism	Feature	Reference
**RA Transport**					
near *TTR*	rs1667255	0.50	The SNP has been associated with circulating retinol levels	Downstream of gene	[38,39]
*FFAR4*	rs10882272	0.39	-	3′ UTR	[38,39]
**Cleavage β-Carotene into RA**					
near *BCO1*	rs4889286	0.49	Associates with plasma β-carotene	Upstream of gene	[40]
*BCO1*	rs12934922	0.23	-	Missense (Arg to Ser)	[40,41]
near *BCO1*	rs6564851	0.48	-	Upstream of gene	[40]
**RA Receptor**					
*RARB*	rs6800566	0.25	Associated with immune response and/or cytokine levels after stimulation	Intron	[42]
*RARB*	rs13070407	0.20	-	Intron	
**Uptake of β-Carotene into Enterocytes**					
*ABCA1*	rs2791952	0.14	Associated with β-carotene bioavailability	Intron	[43,44]
*FABP2*	rs1799883	0.25	Affect the promotor activity in in vitro promotor assay	Missense (Ala to Thr))	[45]

MAF, minor allele frequencies in the population; RA, retinoic acid; SNP, single nucleotide polymorphism; UTR, un-translated region. Proteins encoded by genes: transthyretin encoded by TTR, free fatty acid receptor 4 endcoded by FFAR4, beta-carotene oxygenase 1 encoded by BCO1, retinoic acid receptor beta encoded by RARB, ATP binding cassette subfamily A member 1 encoded by ABCA1, fatty acid binding protein 2 encoded by FABP2.

**Table 3 nutrients-11-01428-t003:** Incidence rate ratios (IRR) for associations between the polymorphisms and colorectal cancer (CRC).

Polymorphism	*n*_cases_ (%)	*n*_sub-cohort_ (%)	IRR (95% CI) ^1^	IRR (95% CI) ^2^	*p*-Value ^3^
*TTR* rs1667255					
AA	360 (39)	675 (39)	1.00 (ref.)	1.00 (ref.)	
CA	426 (46)	794 (46)	0.99 (0.83–1.19)	1.03 (0.86–1.23)	0.76
CC	139 (15)	270 (16)	0.97 (0.76–1.24)	0.99 (0.77–1.27)	0.93
CA + CC	565 (61)	1064 (61)	0.99 (0.84–1.17)	1.02 (0.86–1.21)	0.84
CC vs. AA + CA	139 (15)	270 (16)	0.97 (0.78–1.22)	0.97 (0.78–1.22)	0.82
*FFAR4* rs10882272					
TT	395 (43)	699 (40)	1.00 (ref.)	1.00 (ref.)	
TC	414 (45)	811 (47)	0.92 (0.77–1.10)	0.92 (0.77–1.10)	0.35
CC	116 (13)	232 (13)	0.90 (0.69–1.16)	0.93 (0.71–1.21)	0.57
TC + CC	530 (57)	1043 (60)	0.91 (0.78–1.08)	0.92 (0.78–1.09)	0.33
*BCO1* rs4889286					
TT	250 (27)	451 (26)	1.00 (ref.)	1.00 (ref.)	
TC	451 (48)	862 (49)	0.95(0.78–1.15)	0.96 (0.79–1.17)	0.70
CC	237 (25)	446 (25)	0.97(0.77–1.21)	0.99 (0.79–1.25)	0.97
TC + CC	688 (73)	1308 (74)	0.96(0.80–1.15)	0.97 (0.81–1.17)	0.77
*BCO1* rs12934922					
AA	274 (30)	541 (31)	1.00 (ref.)	1.00 (ref.)	
TA	437 (47)	837 (48)	1.05 (0.87–1.27)	1.05 (0.87–1.28)	0.61
TT	211 (23)	364 (21)	1.18 (0.94–1.48)	1.18 (0.93–1.48)	0.18
TA + TT	648 (70)	1201 (69)	1.09 (0.91–1.30)	1.09 (0.91–1.31)	0.36
*BCO1* rs6564851					
GG	252 (27)	467 (27)	1.00 (ref.)	1.00 (ref.)	
TG	444 (48)	854 (49)	0.96 (0.79–1.17)	0.98 (0.80–1.19)	0.82
TT	230 (25)	433 (25)	1.00 (0.80–1.25)	1.03 (0.82–1.30)	0.80
TG + TT	674 (73)	1287 (73)	0.97 (0.81–1.17)	0.99 (0.82–1.20)	0.96
TT vs. GG + TG	230 (25)	433 (25)	1.02 (0.85–1.24)	1.05 (0.86–1.27)	0.65
*RARB* rs6800566					
GG	397 (43)	735 (42)	1.00 (ref.)	1.00 (ref.)	
GA	421 (45)	830 (47)	0.93 (0.78–1.11)	0.94 (0.79–1.12)	0.51
AA	108 (12)	199 (11)	1.00 (0.76–1.31)	0.98 (0.75–1.30)	0.92
GA + AA	529 (57)	1029 (58)	0.94 (0.80–1.11)	0.95 (0.80–1.12)	0.56
*RARB* rs13070407					
TT	536 (57)	985 (56)	1.00 (ref.)	1.00 (ref.)	
TC	344 (37)	672 (38)	0.96 (0.81–1.14)	0.95 (0.80–1.14)	0.61
CC	60 (6)	110 (6)	1.04 (0.74–1.46)	1.06 (0.75–1.50)	0.75
TC + CC	404 (43)	782 (44)	0.97 (0.83–1.15)	0.97 (0.82–1.14)	0.71
CC vs. TT + TC	60 (6)	110 (6)	1.05 (0.75–1.47)	1.08 (0.77–1.51)	0.66
*ABCA1* rs2791952					
CC	720 (77)	1389 (79)	1.00 (ref.)	1.00 (ref.)	
TC	211 (22)	341 (19)	1.17 (0.96–1.43)	1.15 (0.93–1.40)	0.19
TT	7 (1)	27 (2)	0.48 (0.21–1.10)	0.51 (0.22–1.19)	0.13
TC + TT	218 (23)	368 (21)	1.12 (0.92–1.36)	1.10 (0.90–1.35)	0.34
*FABP2* rs1799883					
GG	505 (55)	908 (52)	1.00 (ref.)	1.00 (ref.)	
GA	356 (39)	706 (41)	0.89 (0.75–1.05)	0.89 (0.74–1.06)	0.18
AA	62 (7)	125 (7)	0.89 (0.64–1.24)	0.88 (0.63–1.23)	0.45
GA + AA	418 (45)	831 (48)	0.89 (0.75–1.04)	0.88 (0.75–1.05)	0.16

^1^ IRRs for CRC estimated by the Cox proportional hazards model with age as the underlying time axis, and stratified by gender, so that the underlying baseline hazards are gender specific. 95% CI is based on Wald’s tests. ^2^ In addition, adjusted for smoking status, alcohol, HRT status (women only), BMI, use of NSAID, energy consumption, intake of red and processed meat dietary fiber, fruit and vegetable intake. ^3^
*p*-value for adjusted risk estimates. Number of missing observations; *TTR* rs1667255 230, *FFAR4* rs10882272 227, *BCO1* rs4889286 197, *BCO1* rs12934922 230, *BCO1* rs6564851 214, *RARB* rs6800566 203, *RARB* rs13070407 186, *ABCA1* rs2791952 199, *FABP2* rs1799883 232.

**Table 4 nutrients-11-01428-t004:** Interactions between polymorphisms and dietary intake of vitamin A, β-carotene, and retinol and risk of colorectal cancer.

	Vitamin A		β-Carotene		Retinol	
	IRR (95% CI) ^1^	*p*-Value	IRR (95% CI) ^1^	*p*-Value	IRR (95% CI) ^1^	*p*-Value
*TTR* rs1667255						
AA	0.98 (0.88–1.09)	0.61	1.02 (0.89–1.17)	0.92	0.99 (0.83–1.19)	0.32
CA + CC	1.01 (0.91–1.13)		1.03 (0.91–1.17)		1.11 (0.94–1.32)	
*FFAR4* rs10882272						
TT	1.05 (0.94–1.17)	0.25	1.05 (0.93–1.20)	0.45	1.16 (0.97–1.38)	0.21
TC + CC	0.97 (0.87–1.07)		1.00 (0.88–1.13)		1.01 (0.85–1.19)	
*BCO1* rs4889286						
TT	1.07 (0.91–1.26)	0.33	1.20 (0.97–1.49)	0.08	1.04 (0.83–1.31)	0.85
TC + CC	0.98 (0.90–1.08)		1.00 (0.90–1.12)		1.07 (0.92–1.24)	
*BCO1* rs12934922						
AA	1.00 (0.86–1.15)	0.92	0.94 (0.79–1.13)	0.20	1.16 (0.96–1.40)	0.21
TA + TT	1.00 (0.91–1.10)		1.06 (0.94–1.20)		1.00 (0.85–1.18)	
*BCO1* rs6564851						
GG	1.06 (0.90–1.25)	0.37	1.21 (0.98–1.50)	0.09	1.02 (0.81–1.28)	0.77
TG + TT	0.98 (0.90–1.08)		1.02 (0.91–1.14)		1.06 (0.91–1.23)	
*RARB* rs6800566						
GG	0.98 (0.88–1.10)	0.53	0.97 (0.85–1.12)	0.23	1.10 (0.92–1.31)	0.61
GA + AA	1.03 (0.92–1.14)		1.06 (0.94–1.20)		1.04 (0.88–1.22)	
*RARB* rs13070407						
TT	1.02 (0.92–1.13)	0.46	1.08 (0.96–1.22)	0.05	1.01 (0.86–1.19)	0.35
TC + CC	0.97 (0.86–1.09)		0.92 (0.79–1.08)		1.13 (0.93–1.36)	
*ABCA1* rs2791952						
CC	0.98 (0.90–1.08)	0.43	0.99 (0.88–1.12)	0.52	1.09 (0.95–1.27)	0.47
TC + TT	1.05 (0.90–1.22)		1.04 (0.91–1.21)		1.00 (0.80–1.25)	
*FABP2* rs1799883						
GG	1.00 (0.89–1.11)	0.73	1.04 (0.92–1.18)	0.67	1.09 (0.93–1.29)	0.35
GA + AA	0.97 (0.87–1.09)		1.01 (0.87–1.16)		0.98 (0.82–1.18)	

Risk estimate for the increment of 1000 retinol equivalents (RE) per day. IRR, incidence rate ratio; CI, confidence interval (Vitamin A: Risk estimate for the increment of 1000 retinol equivalents (RE) per day, β-Carotene: Risk estimate for the increment of 4000 μg per day, Retinol Risk estimate for the increment of 1000 μg per day). ^1^
*p*-value for interaction between genotype and intake of vitamin A, β-carotene or retinol for adjusted risk estimates. Number of missing observations; *TTR* rs16672552 294, *FFAR4* rs10882272 292, *BCO1* rs4889286 263, *BCO1* rs12934922 291, *BCO1* rs6564851 279, *RARB* rs6800566 266, *RARB* rs13070407 251, *ABCA1* rs2791952 264, *FABP2* rs1799883 297.

**Table 5 nutrients-11-01428-t005:** Tertile analyses of polymorphisms and dietary intake of vitamin A, β-carotene, and retinol.

	1. Tertile	2. Tertile	3. Tertile	1. Tertile	2. Tertile	3. Tertile	*p*	1. Tertile	2. Tertile	3. Tertile	1. Tertile	2. Tertile	3. Tertile	*p*
	Nc/Ns	Nc/Ns	Nc/Ns	IRR (95% CI)	IRR (95% CI)	IRR (95% CI)		Nc/Ns	Nc/Ns	Nc/Ns	IRR (95% CI)	IRR (95% CI)	IRR (95% CI)	
*TTR* rs1667255				Vitamin A							Retinol			
AA	119/216	126/228	105/219	1.00	1.02 (0.66–1.57)	0.93 (0.58–1.49)		110/221	127/237	113/205	1.00	1.02 (0.66–1.58)	1.02 (0.63–1.63)	
CA + CC	185/335	178/340	190/359	1.03 (0.70–1.53)	0.99 (0.67–1.48)	1.04 (0.68–1.59)	0.83	162/355	178/320	213/359	0.93 (0.62–1.38)	1.08 (0.72–1.62)	1.14 (0.74–1.76)	0.66
				β-carotene										
AA	131/199	118/229	101/235	1.00	0.90 (0.57–1.42)	0.77 (0.46–1.29)								
CA + CC	196/327	196/348	161/359	0.95 (0.65–1.40)	0.98 (0.65–1.49)	0.83 (0.51–1.36)	0.79							
*FFAR4* rs10882272				Vitamin A							Retinol			
TT	129/236	134/230	125/213	1.00	1.16 (0.76–1.76)	1.19 (0.76–1.87)		118/231	128/244	142/204	1.00	0.97 (0.64–1.49)	1.31 (0.83–2.06)	
TC + CC	172/317	168/341	174/363	1.07 (0.73–1.58)	0.95 (0.64–1.41)	0.98 (0.65–1.48)	0.36	154/344	175/314	185/363	0.90 (0.60–1.34)	1.07 (0.71–1.59)	0.97 (0.63–1.48)	0.18
				β-carotene										
TT	139/217	140/238	109/224	1.00	1.09 (0.71–1.68)	0.90 (0.54–1.50)								
TC + CC	187/311	172/340	155/370	0.99 (0.68–1.45)	0.90 (0.59–1.36)	0.79 (0.49–1.28)	0.67							
*BCO1* rs4889286				Vitamin A							Retinol			
TT	78/162	79/140	83/142	1.00	1.12 (0.66–1.89)	1.22 (0.71–2.09)		69/145	91/163	80/136	1.00	1.06 (0.63–1.78)	1.10 (0.62–1.95)	
TC + CC	230/397	227/437	217/439	1.18 (0.77–1.80)	1.09 (0.71–1.68)	1.08 (0.68–1.70)	0.48	206/438	217/399	251/436	0.98 (0.62–1.53)	1.08 (0.69–1.70)	1.12 (0.70–1.79)	0.98
				β-carotene										
TT	77/151	87/145	76/148	1.00	1.27 (0.75–2.16)	1.14 (0.63–2.04)								
TC + CC	253/380	233/440	188/453	1.31 (0.86–2.01)	1.16 (0.73–1.82)	0.93 (0.55–1.57)	0.12							
*BCO1* rs12934922				Vitamin A							Retinol			
AA	89/178	98/155	85/192	1.00	1.23 (0.75–2.01)	0.96 (0.58–1.59)		73/181	91/187	108/157	1.00	1.09 (0.65–1.81)	1.61 (0.95–2.75)	
TA + TT	214/376	203/414	212/387	1.18 (0.78–1.79)	1.04 (0.69–1.58)	1.16 (0.74–1.79)	0.23	200/395	209/371	220/411	1.27 (0.82–1.96)	1.38 (0.88–2.17)	1.25 (0.78–2.00)	0.05
				β-carotene										
AA	94/154	103/170	75/201	1.00	1.09 (0.66–1.80)	0.70 (0.39–1.24)								
TA + TT	230/374	212/409	187/394	1.05 (0.69–1.59)	0.98 (0.63–1.53)	0.91 (0.55–1.51)	0.27							
*BCO1* rs6564851				Vitamin A							Retinol			
GG	79/168	81/145	82/146	1.00	1.11 (0.66–1.87)	1.20 (0.70–2.05)		73/153	91/166	78/140	1.00	1.06 (0.63–1.77)	1.08 (0.61–1.90)	
TG + TT	224/388	222/431	215/434	1.19 (0.78–1.82)	1.11 (0.72–1.69)	1.09 (0.69–1.72)	0.53	199/426	215/392	247/435	0.98 (0.63–1.52)	1.11 (0.71–1.73)	1.13 (0.71–1.80)	0.95
				β-carotene										
GG	78/155	85/151	79/153	1.00	1.23 (0.72–2.09)	1.14 (0.64–2.05)								
TG + TT	248/379	225/432	188/442	1.30 (0.85–1.99)	1.15 (0.73–1.82)	0.98 (0.58–1.66)	0.19							
*RARB* rs6800566				Vitamin A							Retinol			
GG	132/239	133/246	123/232	1.00	0.98 (0.65–1.48)	1.04 (0.66–1.63)		119/250	138/239	131/228	1.00	1.19 (0.78–1.82)	1.14 (0.73–1.80)	
GA + AA	171/325	170/330	176/350	0.95 (0.65–1.40)	0.96 (0.65–1.43)	0.96 (0.64–1.45)	0.96	155/334	165/326	197/345	1.00 (0.67–1.49)	1.01 (0.68–1.51)	1.16 (0.76–1.77)	0.63
				β-carotene										
GG	142/229	142/241	104/247	1.00	1.08 (0.70–1.67)	0.81 (0.48–1.36)								
GA + AA	185/305	171/346	161/354	0.99 (0.68–1.46)	0.91 (0.60–1.36)	0.87 (0.54–1.38)	0.51							
*RARB* rs13070407				Vitamin A							Retinol			
TT	168/305	179/313	177/348	1.00	1.10 (0.76–1.60)	1.01 (0.68–1.49)		161/315	166/323	197/328	1.00	0.97 (0.67–1.41)	1.12 (0.74–1.67)	
TC + CC	141/257	129/264	122/239	1.08 (0.74–1.59)	0.92 (0.62–1.37)	1.02 (0.66–1.58)	0.46	115/270	144/243	133/247	0.87 (0.58–1.30)	1.11 (0.75–1.65)	1.01 (0.66–1.54)	0.38
				β-carotene										
TT	184/287	167/332	173/347	1.00	0.90 (0.60–1.33)	0.93 (0.59–1.48)								
TC + CC	147/247	152/255	93/258	0.99 (0.68–1.45)	1.08 (0.72–1.62)	0.67 (0.40–1.13)	0.06							
*ABCA1* rs2791952				Vitamin A							Retinol			
CC	237/435	237/452	227/467	1.00	0.99 (0.72–1.36)	0.97 (0.68–1.37)		207/456	242/444	252/454	1.00	1.18 (0.85–1.63)	1.19 (0.84–1.70)	
TC + TT	69/124	71/124	74/113	1.03 (0.64–1.64)	1.06 (0.66–1.68)	1.18 (0.72–1.95)	0.78	68/126	67/118	79/117	1.28 (0.81–2.04)	1.13 (0.69–1.84)	1.33 (0.81–2.21)	0.52
				β-carotene										
CC	251/421	250/446	200/487	1.00	1.03 (0.74–1.46)	0.79 (0.52–1.20)								
TC + TT	78/114	69/131	67/116	1.06 (0.67–1.68)	0.96 (0.60–1.55)	1.11 (0.64–1.93)	0.24							
*FABP2* rs1799883				Vitamin A							Retinol			
GG	159/284	167/312	165/296	1.00	0.94 (0.65–1.37)	1.03 (0.69–1.55)		154/300	154/287	183/305	1.00	0.97 (0.66–1.43)	1.10 (0.73–1.65)	
GA + AA	146/266	133/257	128/284	0.93 (0.63–1.37)	0.92 (0.62–1.38)	0.82 (0.53–1.25)	0.57	118/267	147/273	142/267	0.85 (0.57–1.27)	0.98 (0.66–1.45)	0.94 (0.61–1.45)	0.68
				β-carotene										
GG	183/300	161/288	147/304	1.00	1.04 (0.70–1.53)	0.91 (0.57–1.45)								
GA + AA	144/229	148/287	115/291	1.01 (0.69–1.49)	0.90 (0.60–1.35)	0.74 (0.46–1.20)	0.58							

Nc, Ncases; Ns, Nsubcohort. Number of missing observations; *TTR* rs1667255 294, *FFAR4* rs10882272 292, *BCO1* rs4889286 263, *BCO1* rs12934922 291, *BCO1* rs6564851 279, *RARB* rs6800566 266, *RARB* rs13070407 251, *ABCA1* rs2791952 264, *FABP2* rs1799883 297. Vitamin A 1st tertile 300–1600 RE/day, 2nd tertile 1600–2500 RE/day, 3rd tertile 2500–15,500 RE/day. Retinol 1st tertile 90–730 μg/day, 2nd tertile 730–1280 μg/day, 3rd tertile 1280–5660 μg/day. β-carotene l 1st tertile 160–2180 μg/day, 2nd tertile 2180–4710 μg/day, 3rd tertile 4710–54,180 μg/day.

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
