# Peer review of "No Interaction between Polymorphisms Related to Vitamin A Metabolism and Vitamin A Intake in Relation to Colorectal Cancer in a Prospective Danish Cohort"

_nutrients, 2019, doi:10.3390/nu11061428_

Round 1

Reviewer 1 Report

The manuscript by Andersen et al. represents an important contribution to the field, as it clarifies the relationship between vitamin A intake and signaling and colorectal cancer. Although overall informative, the presentation of the data has major deficiencies that need to be addressed before publication. 

1. The title is too long and too confusing. One simpler, alternative tile is listed below:

There is no association between the risk of colorectal cancer and vitamin A intake, metabolism or signaling; perspective cohort studies in the Danish population

2. The results are a conglomeration of tables without proper description (except Table 4). This defeats the purpose of the results section, which in addition to the presentation of the data should provide a thorough description/interpretation.

3. There are several scientific inaccuracies that need to corrected:

- line 47 – ‘…, vitamin A (including retinal, retinol, retinoic acid, ….). Vitamin A is defined as all-trans-retinol not as any other metabolite. The word ‘vitamin A’ should be replaced by ‘retinoids’, which more accurately describe the group of compounds.

-line 49 –‘ Vitamin A from food derives mainly from fat-soluble retinol from …’. Vitamin A is a retinol and cannot be derived from it. The main sources of preformed vitamin A are retinyl esters. 

Line 284 – ‘…no support for strong involvement…’. The lack of ‘strong involvement” still implies some level of weak involvement.  However, as stated in the abstract and the presented data there was absolutely no association between vitamin A and colorectal cancer. Please make the final message cohesive.

4. Editorial and grammar insufficiencies: 

In general the manuscript needs to be proofread by a native English speaker. There are numerous awkward grammar constructions and mistakes. It would take much too long to list them all. 

-line 57 – ‘Very interesting,…’ Should be just ‘Interestingly’. This allows the readers decide the level of their interest.  

-line 86 – sentence starting with ‘Vitamin A intake ….’ does not make sense.

-line 87 – sentence starting with ‘Additionally, ….’ Is much too long and needs to be revised. 

-line 93 – ‘We utilized…’. Should be ‘We assumed…” Anyway the whole sentence is cut in half (ends in line 94 without logical continuation). 

etc.

Author Response

All reviewers essentially addressed the same issues. We have therefore decided to send all the comments and the responses to all reviewers. We hope that you can accept this approach.

On behalf of all the authors  Vibeke Andersen

Reviewer 1:

The manuscript by Andersen et al. represents an important contribution to the field, as it clarifies the relationship between vitamin A intake and signaling and colorectal cancer. Although overall informative, the presentation of the data has major deficiencies that need to be addressed before publication. 

1. The title is too long and too confusing. One simpler, alternative tile is listed below:

There is no association between the risk of colorectal cancer and vitamin A intake, metabolism or signaling; perspective cohort studies in the Danish population

2. The results are a conglomeration of tables without proper description (except Table 4). This defeats the purpose of the results section, which in addition to the presentation of the data should provide a thorough description/interpretation.

3. There are several scientific inaccuracies that need to corrected:

- line 47 – ‘…, vitamin A (including retinal, retinol, retinoic acid, ….). Vitamin A is defined as all-trans-retinol not as any other metabolite. The word ‘vitamin A’ should be replaced by ‘retinoids’, which more accurately describe the group of compounds.

-line 49 –‘ Vitamin A from food derives mainly from fat-soluble retinol from …’. Vitamin A is a retinol and cannot be derived from it. The main sources of preformed vitamin A are retinyl esters. 

Line 284 – ‘…no support for strong involvement…’. The lack of ‘strong involvement” still implies some level of weak involvement.  However, as stated in the abstract and the presented data there was absolutely no association between vitamin A and colorectal cancer. Please make the final message cohesive.

4. Editorial and grammar insufficiencies: 

In general the manuscript needs to be proofread by a native English speaker. There are numerous awkward grammar constructions and mistakes. It would take much too long to list them all. 

-line 57 – ‘Very interesting,…’ Should be just ‘Interestingly’. This allows the readers decide the level of their interest.  

-line 86 – sentence starting with ‘Vitamin A intake ….’ does not make sense.

-line 87 – sentence starting with ‘Additionally, ….’ Is much too long and needs to be revised. 

-line 93 – ‘We utilized…’. Should be ‘We assumed…” Anyway the whole sentence is cut in half (ends in line 94 without logical continuation). 

etc.

RE:

We are very grateful  to the reviewer for contributing to improve our manuscript.

1) The title has been shortened to read ” No Interaction between Polymorphisms related to Vitamin A Metabolism and Vitamin A Intake in relation to Colorectal Cancer in a prospective Danish cohort”

2) We agree that the result section is very short, but that is due to all the null findings. However, a more thorough description of the results have now been provided for most of the Tables. For Table 3, where there is no positive association, the short description has been kept. For Table 1, the results have been published previously except for vitamin A intakes for which there were no associations with CRC risk.

3) Line 47. In general, it is acknowledged that vitamin A has three active forms (retinal, retinol and retinoic acid) and a storage form (retinyl ester – see references below). It is from a nutritional point of view not strictly defined as only all-trans-retinol. Retinoids on the other hand can include many more compounds, i.e. also those with no vitamin A activity (Clin Interv Aging. 2006 Dec; 1(4): 327–348.). Thus, though we agree that definitions could vary, we would prefer to stick to vitamin A and describe what is meant be this.

Various definitions for Vitamin A:

WHO. FAO. Vitamin and Mineral Requirements in Human Nutrition. World Health Organization, Food and Agricultural Organization of the United Nations; Geneva, Switzerland: 2004.

PubMed MESH definition: Dietary vitamin A is derived from a variety of CAROTENOIDS found in plants. It is enriched in the liver, egg yolks, and the fat component of dairy products.

Wikipedia definition: Vitamin A is a group of unsaturated nutritional organic compounds that includes retinol, retinal, retinoic acid, and several provitamin A carotenoids (most notably beta-carotene). ("Vitamin A". Micronutrient Information Center, Linus Pauling Institute, Oregon State University, Corvallis. January 2015. Retrieved 6 July 2017, Fennema O (2008). Fennema's Food Chemistry. CRC Press Taylor & Francis. pp. 454–455. ISBN 9780849392726).

Line 49. We have rephrased to read: “Vitamin A sources from food include mainly fat-soluble retinol from animal sources (e.g. liver) and green-or yellow-coloured carotenoids from vegetables (e.g. leafy vegetables, carrots)”.

Line 284. We agree and now write: no support of an involvement of vitamin A in CRC aetiology.

4) The manuscript has now been proofread by a native English speaker.

-line 57 – ‘Very interesting,…’ has been changed to ‘Interestingly’.

-line 86 –94 has been rephrased to read: ” Vitamin A intake was analysed as total intake (dietary and supplements combined), as dietary intake (retinol and β-carotene as a weighted sum), and supplements separately. We also evaluated potential effects of retinol (mainly from meat and meat products) and β-carotene (mainly from plant foods) separately. We selected functional variants in genes involved in vitamin A metabolism.

.  

Reviewer 2:

The authors using functional polymorphisms, gene-environment (GxE) interaction analyses to evaluate potential interactions between vitamin A intake and genetic variation in the A vitamin metabolic pathway genes related to risk of CRC. The results show negative association among these events.

The major concern is the genes selected by authors, the only several genes limit this present study and the conclusion. No significant finding for each of these genes, the author should add at least 1-2 genes (may be in other pathway) that are with positive association compared to selected genes in present data.  

Minor:  is it possible to short the long title?  

             Dark background in figure 1, please modify.

RE:

1) Thanks to the reviewer for this comment. We agree that it will be valuable to include genes for which a positive association is found. However, this manuscript deals with the Vitamin A pathways where we did not find any positive associations. We are concerned about including polymorphisms from another pathway because that might change the focus of the current analysis. We therefore choose to add to the discussion that this cohort has already proven to be valuable for identification of GxE interactions. “This design has previously proven its value by the identification of GxE interactions between e.g. meat and pattern recognition receptors.”

2) The Figure 1 has been modified accordingly.

Reviewer 3.

Remarks to the Author:
1. The title is somewhat diffuse, could you please rewrite it reflecting the main finding of the work.
2.There were many errors in the English including spelling and grammar, which must be improved
3.Rewrite the introduction and discussion to make the findings more appealing for a reader. and also includes suggested reference in the introduction.
4.In line 62-63. of introduction, Please include this reference "Nrf2 driven TERT regulates pentose phosphate pathway in glioblastoma"

5.English language changes in line 29, 34,40-41, 58, 154,156, 166, 169.

6. English Spelling mistakes 216, 222

7.English changes suggested in line 214: Persons to people, line 230: Intakes to Intake, line 240: Includes to Included, line 253: Based on their.

8. English changes in line 272 and 248.

RE:

1. We have changed the title accordingly.

2. The manuscript has now been proofread by a native English speaker.

3. We have been rewritten the introduction and discussion in order to make it more appealing.

4. We have now included the reference (Ahmad F, Dixit D, Sharma V, et al. Nrf2-driven TERT regulates pentose phosphate pathway in glioblastoma. Cell death & disease 2016; 7: e2213).

5. Thank you for helping with the language. Unfortunately, the numbers doesn’t seem to correspond to the numbers in the current manuscript (the one we received from the journal). We have, however, done our best to improve the English language.

Reviewer 2 Report

The authors using functional polymorphisms, gene-environment (GxE) interaction analyses to evaluate potential interactions between vitamin A intake and genetic variation in the A vitamin metabolic pathway genes related to risk of CRC. The results show negative association among these events.

The major concern is the genes selected by authors, the only several genes limit this present study and the conclusion. No significant finding for each of these genes, the author should add at least 1-2 genes (may be in other pathway) that are with positive association compared to selected genes in present data.  

Minor:  is it possible to short the long title?  

             Dark background in figure 1, please modify.

Author Response

(The authors gave the same response as above.)

Reviewer 3 Report

Remarks to the Author:
1. The title is somewhat diffuse, could you please rewrite it reflecting the main finding of the work.
2.There were many errors in the English including spelling and grammar, which must be improved
3.Rewrite the introduction and discussion to make the findings more appealing for a reader. and also includes suggested reference in the introduction.
4.In line 62-63. of introduction, Please include this reference "Nrf2 driven TERT regulates pentose phosphate pathway in glioblastoma"

5.English language changes in line 29, 34,40-41, 58, 154,156, 166, 169.

6. English Spelling mistakes 216, 222

7.English changes suggested in line 214: Persons to people, line 230: Intakes to Intake, line 240: Includes to Included, line 253: Based on their.

8. English changes in line 272 and 248.

Author Response

(The authors gave the same response as above.)

Round 2

Reviewer 1 Report

I do not have additional comments to the manuscript. 

Reviewer 2 Report

Accept this revised version, thanks